# Association of body mass index and waist circumference with long-term mortality risk in 10,370 coronary patients and potential modification by lifestyle and health determinants

**Esther Cruijsen**[1]*, **Nadia E. Bonekamp**[2], **Charlotte Koopal**[2], **Renate M. Winkels**[1], **Frank L. J. Visseren**[2], **Johanna M. Geleijnse**[1], **on behalf of the Alpha Omega Cohort study group and the UCC-SMART study group**[¶]

1 Division of Human Nutrition and Health, Wageningen University & Research, Wageningen, The Netherlands, 2 Department of Vascular Medicine, University Medical Center Utrecht, Utrecht University, Utrecht, The Netherlands

¶ Membership of the Alpha Omega Cohort study group and the UCC-SMART study group is provided in the Acknowledgments.

* esther.cruijsen@wur.nl

## Abstract

### Background and aims

Body adiposity is known to affect mortality risk in patients with coronary artery disease (CAD). We examined associations of body mass index (BMI) and waist circumference (WC) with long term mortality in Dutch CAD patients, and potential and effect modification of these associations by lifestyle and health determinants.

### Methods

10,370 CAD patients (mean age ∼65 y; 20% female; >80% on cardiovascular drugs) from the prospective Alpha Omega Cohort and Utrecht Cardiovascular Cohort–Secondary Manifestations of ARTerial disease study were included. Cox models were used to estimate categorical and continuous associations (using restricted cubic splines) of measured BMI and WC with all-cause and cardiovascular mortality risk, adjusting for age, sex, smoking, alcohol, physical activity and educational level. Analyses were repeated in subgroups of lifestyle factors (smoking, physical activity, diet quality), education and health determinants (diabetes, self-rated health).

### Results

During ∼10 years of follow-up (91,947 person-years), 3,553 deaths occurred, including 1,620 from cardiovascular disease. U-shaped relationships were found for BMI and mortality risk, with the lowest risk for overweight patients (BMI ∼27 kg/m²). For obesity (BMI ≥30), the HR for all-cause mortality was 1.31 (95% CI: 1.11, 1.41) in male patients and 1.10 (95%

**Data Availability Statement:** For data from the Alpha Omega Cohort: Metadata, a read-me file, and codebook are available at DOI: 10.4121/0b66dd27-ee11-43f2-b58b-fff98463f944 (Wageningen University & Research). The repository does not contain the minimal dataset to reproduce the results due to ethical restrictions as participants did not consent to sharing their data publicly. The minimal dataset is secured at the Division of Human Nutrition and Health, Wageningen University and Research, Wageningen, The Netherlands. The data stewards at the Wageningen University are responsible for research data management and reviewing requests to access the minimal dataset. For data from the UCC-SMART cohort: Data cannot be shared publicly as stated in the protocol approved by the Medical Research Ethics Committee (MREC) NedMec of the UMC Utrecht (reference 13-597). Data are available upon reasonable request. The UCC-SMART Study group directs the academic focus of research using the UCC-SMART data and consists of members from both epidemiological and clinical cardiovascular research. Datasets are provided to interested researchers after approval of request by the UCC-SMART Study group. Access to the data request module can be applied for via ucc-smart@umcutrecht.nl.

**Funding:** This work was supported by a grant from the Regio Deal Foodvalley (162135). The Alpha Omega Trial (2002–2009), from which this cohort study emerged, was supported by Netherlands Heart Foundation grant 200T401; NIH, National Heart, Lung, and Blood Institute, and Office of Dietary Supplements grant R01HL076200.

**Competing interests:** The authors have declared that no competing interests exist.

CI: 0.92, 1.30) in female patients, compared to BMI 25–30 kg/m$^2$. WC was also non-linearly associated with mortality, and HRs were 1.18 (95%CI:1.06, 1.30) in males and 1.31 (95% CI:1.05, 1.64) in females for the highest vs. middle category of WC. Results for cardiovascular mortality were mostly in line with the results for all-cause mortality. U-shaped associations were found in most subgroups, associations were moderately modified by physical activity, smoking and educational level.

## Conclusions

CAD patients with obesity and a large WC were at increased risk of long-term CVD and all-cause mortality, while mildly overweight patients had the lowest risk. These associations were consistent across subgroups of patients with different lifestyles and health status.

## Introduction

Cardiovascular risk management includes lifestyle modifications such as smoking cessation, physical activity and adopting a healthy diet, in addition to pharmacological treatment [1]. Maintaining a body mass index (BMI) between 18.5 and 25.0 kg/m$^2$ and a waist circumference (WC) <94 cm for males and <80 cm for females is considered important for cardiovascular health in the general population [2]. In cardiovascular disease (CVD) patients clinical guidance on weight loss targets is complicated, because J- or U- shaped relationships of BMI and WC with mortality have repeatedly been reported [3]. In a meta-analysis of 89 observational studies with over 1.3 million coronary artery disease (CAD) patients, lower risks of long term (≥6 months) all-cause mortality (relative risk [RR]: 0.78, 95% CI: 0.74, 0.82) and CVD mortality (RR: 0.82, 95% CI: 0.74, 0.90) were found for overweight patients (BMI ≥25–30 kg/m$^2$) compared to those with a BMI between ≥18.5–25 kg/m$^2$ [3]. Higher risks of mortality were observed in this meta-analysis for both underweight patients and those with a BMI ≥30 kg/m$^2$ [3]. The reasons behind this "obesity paradox" are not yet fully understood. Possibly overweight CAD patients may have better metabolic reserves to manage their disease. Reverse causality may also play a role, as CAD patients with lower body weight may be more frail due to underlying disease that can cause greater mortality risks. Adiposity, reflected in a higher BMI and WC, is correlated with lower socioeconomic position (SEP) [4–6] and is determined by lifestyle-related behaviours including diet [7], physical activity [8] and smoking [9]. Apart from caloric intake from the diet, the quality of the diet is also important in cardiovascular risk management [10, 11]. A high-quality diet includes a variety of nutrient-dense foods such as fruits, vegetables, whole grains, lean protein sources, and healthy fats, while limiting the intake of saturated fats, added sugars, and sodium [12]. To what extent upstream (i.e. modifiable) lifestyle and health determinants could explain or modify J- or U-shaped relationships of BMI and WC with mortality risk in CAD patients is not yet clear. The nature of the relationship may differ between male and female patients [13]. Risk stratification for these determinants could provide insight in the role of modifying factors and guide health care professionals in identifying patients for weight loss therapy and personalised counselling in weight management. In the present pooling study of two prospective cohorts of CAD patients in the Netherlands, we performed sex-specific analyses of BMI and WC in relation to long-term risk of CVD and all-cause mortality. Exploratory analyses were performed to obtain insight in potential effect modification of these associations by lifestyle and health determinants including

smoking, physical activity, diet quality, educational level, diabetes and self-rated health. Insight in subgroup-specific associations can be useful for risk stratification and prediction, and risk communication to patients.

## Methods

### Study design and participants

The Alpha Omega Cohort (AOC) is a prospective cohort study including 4,837 Dutch male and female patients aged 60–80 year with a verified history of myocardial infarction (MI) ≤ 10 years prior to study enrollment between 2002 and 2006. Patients participated in a randomized controlled trial of omega-3 fatty acids during the first 3 years of follow-up which had no effect on major cardiovascular events [14, 15]. Follow-up for cause-specific mortality started in 2002, and is still ongoing.

The Utrecht Cardiovascular Cohort—Secondary Manifestations of ARTerial disease (UCC-SMART) cohort is an ongoing, prospective study with mainly Dutch patients aged 18–79 years referred to the University Medical Center Utrecht, the Netherlands, for management of cardiovascular disease. The design and rationale of this study have been previously described [16, 17]. For the current analysis, data was used from 5,533 patients with a clinical diagnosis of CAD upon inclusion in the study (defined as MI, percutaneous coronary intervention, coronary artery bypass grafting and angina pectoris) included between 1996 and 2020.

Patients from both cohorts provided written informed consent and the studies were approved by a central ethics committee (AOC) and the ethics committees from participating hospitals (AOC & UCC-SMART).

### Baseline measurements

In both studies, body weight and height were measured at baseline by trained staff according to a standardized protocol, with patients in light clothing without shoes. BMI was calculated as body weight (kg) divided by height (m) squared. WC was measured half-way between the lower rib and the iliac crest with patients in standing position and light clothing. In UCC-S-MART, WC was measured twice and averaged, if measurements differed by > 2 cm, a third measurement was taken. BMI and WC were categorized based on general CVD prevention guidelines [1]. For BMI the following categories were used: 1) BMI < 25 kg/m$^2$; 2) BMI ≥ 25–30 kg/m$^2$; 3) BMI ≥ 30 kg/m$^2$. For WC the following sex-specific categories were used: 1) <94 cm for males, <80 cm for females; 2) ≥ 94–102 cm for males; ≥ 80–88 cm for females; 3) ≥ 102 cm for males; ≥ 88 cm for females.

Data on smoking, physical activity, alcohol intake, educational level, medication use and other risk factors were collected in both studies using self-reported questionnaires. Smoking status was assessed in three categories: never, former and current smokers. Educational level was used as proxy for SEP and was divided into four categories: elementary education or less, low education (lower secondary education), moderate education (higher secondary or lower tertiary education) and high education (higher tertiary education) [18]. In AOC, physical activity was assessed by the validated Physical Activity Scale for the Elderly [19] and categorized in four categories according to the Dutch physical activity guidelines: no activity, light (≤ 3 Metabolic Equivalent Tasks (METs)), intermediate (> 3 METs on >0 to <5 days per week) and high activity (> 3 METs on ≥5 days per week). In UCC-SMART, physical activity was assessed by a short version of a validated questionnaire and total amount of physical activity per week in MET hours (METhw) was divided in quartiles (≤ 25, > 25 - ≤ 45, >45 - ≤73 and >73 METhw) [20, 21].

In the AOC, total alcohol intake (g/d) was calculated as the sum of ethanol from all reported alcoholic beverages in a food frequency questionnaire [22], which was classified in sex-specific categories. In men, drinking categories were defined as 0 g/d (abstainers), >0–10 g/d (light), >10–30 g/d (moderate) and >30 g/d (heavy). In women, drinking categories were defined as 0 g/d (abstainers), >0–5 g/d (light), >5–15 g/d (moderate) and >15 g/d (heavy) [23]. In UCC-S-MART, weekly alcohol intake was assessed using a general questionnaire and categorized as 0 drinks / week (none), ≥1–10 drinks / week (light), 11–20 drinks / week (moderate) and >20 drinks / week (heavy).

In the AOC, dietary intake over the past month was assessed with a 203-item validated food frequency questionnaire [22]. Quality of the diet was assessed using the Dutch Healthy Diet–Cardiovascular Disease index (DHD-CVD index) which reflects adherence to the 2023 Dutch dietary guidelines for patients with atherosclerotic CVD as defined by the Dutch Health Council [11]. The DHD-CVD index was created, building upon the 2015 Dutch Healthy Diet index [24] that was constructed from the dietary guidelines for the general population [25]. The maximum score for fish consumption was adjusted to align with the recommendation for CAD patients to consume (any type of) fish 1 to 2 times per week. The maximum score of 10 points was assigned to those consuming at least 1.5 portion of fish per week. Furthermore, the recommendation to consume plant sterol or stanol containing products (e.g. fortified margarines) was included as an additional item, with the maximum sore of 10 points for any intake and 0 points for no intake of these products. The DHD-CVD score consists of 16 items and has a theoretical range of 0 (no adherence) to 160 points (maximal adherence).

In AOC, blood pressure was measured twice using an automatic device (HEM-711; Omron) and averaged. In UCC-SMART, blood pressure was measured twice on both arms in a sitting position using a manual device and averaged. In AOC, serum lipids, plasma glucose and serum high sensitivity C-reactive protein (hs-CRP) were analyzed from non-fasting blood samples using standardized kits and an automated analyzer (Hitachi 912; Roche Diagnostics). In UCC-SMART, they were measured from fasting venous blood using enzymatic kits. In AOC, prevalent diabetes was defined as fasting (>4 hours) plasma glucose ≥ 7.0 mmol/L, non-fasting plasma glucose ≥ 11.1 mmol/L, use of an anti-diabetic medication (Anatomical Therapeutic Chemical (ATC) code A10), or a self-reported physician diagnosis of diabetes. The same definition was used in UCC-SMART, except for a self-reported physician diagnosis.

In AOC, self-rated health was assessed using the question: "How do you rate your overall health at this moment?", using a 5-point answering scale ranging from poor to excellent [26]. Self-reported medication use was checked by research nurses and coded according to the ATC classification system [27].

## Outcome assessment

The primary endpoint of this study was all-cause mortality, CVD mortality was a secondary endpoint. For the AOC, vital status of patients was monitored through linkage with municipal registries, from baseline through December 2018. Follow-up for cause-specific mortality occurred in 3 phases. From 2002–2009 (Alpha Omega Trial), information was obtained from the national mortality registry (Statistics Netherlands (CBS)), treating physicians, and close family members. Primary and contributing causes of death were coded by an independent Endpoint Adjudication Committee, as described previously [15]. After the trial through 2012, data on the primary and contributing causes of death were obtained from CBS only. From 2013 onwards, CBS provided data on the primary cause of death only, and treating physicians were asked to fill out an additional cause of-death questionnaire (response rate: 67%), which was coded by study physicians who were not involved in the current analysis.

The endpoint of CVD mortality was allocated to all patients for whom it was a primary or contributing cause of death, based on any of the data sources. Mortality coding was performed according to the International Classification of Diseases, Tenth Revision [28], where CVD mortality comprised codes I20–I25 (ischemic heart disease), I46 (cardiac arrest), R96 (sudden death, undefined), I50 (heart failure), and I60–I69 (stroke).

In the UCC-SMART study, patients or their relatives received biannual follow-up questionnaires. When an endpoint was reported, additional information was obtained from the treating physician or hospital. Endpoints were independently adjudicated by members of the UCC-SMART study Endpoint Committee in accordance with pre-published definitions [16]. Cardiovascular mortality was defined as sudden death or death from stroke, MI, congestive heart failure, rupture of abdominal aortic aneurysm, or from other causes, e.g. sepsis following stent placement [16].

## Statistical analysis

Baseline characteristics of the AOC and UCC-SMART study are presented for males and females separately as mean ± standard deviation for normally distributed data, median [interquartile range] for skewed variables and number (%) for categorical variables.

Missing data was imputed using single imputation methods with bootstrapping, additive regression and predictive mean matching based on non-missing patient characteristics to minimize loss of statistical power and possible bias. In AOC, missingness was highest for the dietary variables (n = 453, 9%), LDL-cholesterol (n = 345, 7%), hs-CRP (n = 187, 4%) and HDL cholesterol, total cholesterol and triglycerides (n = 131, 3%). In the UCC-SMART study, missingness was highest for educational level (n = 1,746, 32%), WC (n = 410, 7%) and LDL-cholesterol (n = 235, 4%). Missingness for all other variables was <1%.

Cox proportional hazards models were used to estimate hazard ratios (HRs) and 95% confidence intervals (CIs) for CVD mortality and all-cause mortality across categories of BMI and WC, crude and adjusted for confounders in the total population and for males and females separately. HRs in the first adjusted model were adjusted for age and sex, if not used as stratification factor (model 1). Model 2 was additionally adjusted for smoking status (plus number of packyears in UCC-SMART), educational level, alcohol intake and physical activity and considered as the main model for describing the results. Model 3 was additionally adjusted for potential mediating factors: prevalent diabetes, systolic blood pressure, LDL-cholesterol and hs-CRP levels. The proportional hazards assumption was visually checked using the Schoenfeld residuals and fulfilled. Follow-up time was calculated as the period from the date of study enrollment to either the date of death, the date of last contact or the censoring date, which was December 31, 2018 for AOC and December 31, 2019 for UCC-SMART.

HRs were calculated for AOC and UCC-SMART separately and subsequently pooled with a random effects model meta-analysis to obtain pooled HRs.

Restricted cubic splines (RCS) were used to investigate and illustrate non-linear associations for BMI and WC with CVD mortality and all-cause mortality, using model 2. Reference values were 25 kg/m$^2$ for BMI and 94 cm for males and 80 cm for females for WC based on the upper limit of the respective guideline recommendations for BMI and WC. The number of knots was selected based on the Akaike's information criterion (AIC) of the best fitting model and placed on the 10th, 50th and 90th percentile. Outliers were winsorized at the 1st and 99th percentile in the RCS analyses to minimize the impact of outliers, meaning that outliers outside these percentiles were replaced with observations closest to them. Non-linearity of the associations was assessed using the Wald chi-square statistic.

In the subgroup analyses we stratified for smoking (never / former or current), educational level (low [only elementary or low] / high [moderate or high]), diabetes (yes / no) and physical activity level (low [lower two categories] / high [upper two categories]). For AOC, separate subgroup analyses were performed for diet quality (low [< median DHD-CVD index of 88.8] / high [≥median DHD-CVD index of 88.8]) and self-rated health (poor or moderate / good, very good or excellent). All subgroup analyses were also performed in males and females separately. Sensitivity analyses were performed to check for potential reverse causation by excluding the first two and five years of follow-up and excluding patients with cancer at baseline.

Two-sided P values <0.05 were considered statistically significant. All statistical analyses were performed using R version 4.0.2 (R Foundation for Statistical Computing).

## Results

### Baseline characteristics

Table 1 presents baseline characteristics for 4,837 patients of the AOC and 5,533 patients from UCC-SMART, stratified by BMI. Mean ages of patients were 69.0 ± 5.6 years in the AOC and 61.4 ± 9.5 years in the UCC-SMART. In the AOC, 17% currently smoked, this was 28% in UCC-SMART. Use of lipid-lowering and anti-hypertensive medication was >80% for both cohorts.

The mean BMI was 27.5 ± 3.5 kg/m$^2$ for males and 28.6 ± 4.8 kg/m$^2$ for females from the AOC and 27.4 ± 3.8 kg/m$^2$ for males and 27.3 ± 4.7 kg/m$^2$ for females from UCC-SMART. In the AOC, 32% of the patients were obese (BMI > 30 kg/m$^2$) and 22% had a BMI < 25 kg/m$^2$, in UCC-SMART 23% of patients were obese and 28% had a BMI < 25 kg/m$^2$. The mean WC was 103.1 ± 9.8 cm for males and 97.9 ± 11.9 cm for females from the AOC and 98.9 ± 10.7 cm for males and 90.8 ± 12.7 cm for females from UCC-SMART (S1 Table).

### Relation BMI and WC with all-cause and CVD mortality

The median follow-up duration was 12.4 years [8.5–13.8] (53,167 person-years) in the AOC and 9.4 years [4.8–13.7] (38,780 person-years) in UCC-SMART. During the follow-up period, a total of 2,287 deaths (47% of total cohort) occurred of which 1,010 were due to CVD in the AOC and a total of 1,266 deaths (23% of total cohort) occurred in UCC-SMART of which 610 were due to CVD.

Figs 1 and 2 depict continuous associations of BMI and WC with all-cause mortality for males and females from AOC and UCC-SMART using RCS. Significant U-shaped associations for BMI with all-cause mortality were found in males, both in AOC and UCC-SMART. The risk of mortality was lowest for a BMI between 25 and 30 kg/m$^2$, with a nadir of 27.4 (95% CI: 26.8–27.8) kg/m$^2$ and HR of 0.89 (95% CI: 0.83, 0.96) compared to a BMI of 25 kg/m$^2$. Mortality risk in obese male patients (BMI ≥ 30 kg/m$^2$) was 30–40% higher on average, compared to those with BMI of 25 kg/m$^2$. For the smaller group of

female patients, associations were not statistically significant, but suggestive for a lower all-cause mortality risk in those with a BMI of 25–30 kg/m$^2$ compared to those with a BMI of 25 kg/m$^2$. For WC, U-shaped associations for WC with all-cause mortality were found in males with the lowest risk observed at a WC of around 100 cm compared to a WC of 94 cm. In female patients, lowest mortality risks were observed for a WC <80 cm in AOC and a WC 80–90 cm in UCC-SMART compared to a WC of 80 cm. Results for BMI and WC with CVD mortality closely mirrored those for all-cause mortality, especially in males (S1 and S2 Figs). The pattern for CVD mortality showed a linear association for the AOC and a U-shaped association for UCC-SMART.

**Table 1. Baseline characteristics of 10,370 patients in the Alpha Omega Cohort & UCC-SMART, stratified for BMI[1].**

| | Cohort study | | | | | |
|---|---|---|---|---|---|---|
| | Alpha Omega Cohort | | | UCC-SMART cohort | | |
| | (n = 4,837) | | | (n = 5,533) | | |
| | 1 \| BMI < 25 (n = 1,087) | 2 \| BMI ≥ 25–30 (n = 2,575) | 3 \| BMI ≥ 30 (n = 1,175) | 1 \| BMI < 25 (n = 1,576) | 2 \| BMI ≥ 25–30 (n = 2,770) | 3 \| BMI ≥ 30 (n = 1,187) |
| **BMI, kg/m²** | 23.3 ± 1.5 | 27.3 ± 1.4 | 33.0 ± 3.0 | 23.2 ± 1.5 | 27.2 ± 1.4 | 33.0 ± 3.3 |
| **Age** | 69.6 ± 5.6 | 69.1 ± 5.5 | 68.3 ± 5.7 | 62.2 ± 9.9 | 61.6 ± 9.4 | 60.0 ± 9.2 |
| **Males (n, %)** | 850 (78) | 2,135 (83) | 798 (68) | 1,232 (78) | 2,350 (85) | 905 (76) |
| **Smoking** | | | | | | |
| Never | 247 (23) | 389 (15) | 176 (15) | 415 (26) | 649 (23) | 253 (21) |
| Former | 643 (59) | 1,790 (70) | 773 (66) | 752 (48) | 1,507 (54) | 659 (56) |
| Current | 197 (18) | 396 (15) | 226 (19) | 409 (26) | 614 (22) | 275 (23) |
| **Physical activity[2]** | | | | | | |
| Category 1 | 48 (4) | 119 (5) | 109 (9) | 348 (23) | 680 (25) | 357 (31) |
| Category 2 | 403 (37) | 898 (35) | 463 (39) | 366 (24) | 724 (27) | 292 (25) |
| Category 3 | 213 (20) | 534 (21) | 258 (22) | 428 (28) | 675 (25) | 280 (24) |
| Category 4 | 423 (39) | 1,024 (40) | 345 (29) | 376 (25) | 598 (22) | 228 (20) |
| **Alcohol intake[3]** | | | | | | |
| Abstainers | 316 (29) | 721 (28) | 437 (37) | 245 (16) | 432 (16) | 260 (22) |
| Light | 265 (24) | 678 (26) | 293 (25) | 898 (57) | 1,585 (57) | 620 (52) |
| Moderate | 333 (31) | 748 (29) | 274 (23) | 283 (18) | 492 (18) | 201 (17) |
| Heavy | 173 (16) | 428 (17) | 171 (15) | 147 (9) | 252 (9) | 106 (9) |
| **Educational level** | | | | | | |
| Only elementary | 203 (19) | 505 (20) | 295 (25) | 138 (9) | 321 (12) | 195 (16) |
| Low education | 366 (34) | 888 (35) | 458 (39) | 221 (14) | 509 (18) | 232 (20) |
| Moderate education | 343 (32) | 836 (33) | 313 (27) | 626 (40) | 1,215 (44) | 530 (45) |
| High education | 171 (16) | 328 (13) | 98 (8) | 591 (38) | 752 (17) | 230 (19) |
| **Blood pressure, mmHg** | | | | | | |
| Systolic | 139 ± 23 | 142 ± 21 | 142 ± 22 | 134 ± 20 | 137 ± 20 | 139 ± 19 |
| Diastolic | 78 ± 11 | 81 ± 11 | 81 ± 11 | 78 ± 11 | 80 ± 11 | 82 ± 11 |
| **Serum lipids, mmol/L[4]** | | | | | | |
| Total cholesterol | 4.6 [4.0, 5.2] | 4.6 [4.0, 5.3] | 4.7 [4.1, 5.4] | 4.4 [3.9, 5.2] | 4.4 [3.8, 5.2] | 4.4 [3.8, 5.1] |
| LDL cholesterol | 2.5 [2.0, 3.0] | 2.5 [2.0, 3.1] | 2.5 [2.0, 3.0] | 2.5 [2.0, 3.1] | 2.5 [1.9, 3.2] | 2.4 [1.9, 3.0] |
| HDL cholesterol | 1.3 [1.1, 1.6] | 1.2 [1.0, 1.5] | 1.2 [1.0, 1.4] | 1.2 [1.0, 1.5] | 1.1 [1.0, 1.3] | 1.1 [0.9, 1.2] |
| Triglycerides | 1.4 [1.0, 1.9] | 1.7 [1.2, 2.3] | 2.0 [1.5, 2.8] | 1.2 [0.9, 1.6] | 1.4 [1.0, 2.0] | 1.7 [1.2, 2.4] |
| **Hs-CRP, mg/L** | 1.4 [0.7, 3.1] | 1.7 [0.8, 3.5] | 2.6 [1.1, 4.9] | 1.4 [0.7, 3.4] | 1.9 [1.0, 3.7] | 2.6 [1.4, 5.1] |
| **Anti-lipid drug use** | 925 (85) | 2,193 (85) | 1,004 (85) | 1,271 (81) | 2,303 (83) | 992 (84) |
| **Anti-hypertensive drug use** | 938 (86) | 2,306 (90) | 1,096 (93) | 1,374 (87) | 2,510 (91) | 1,106 (93) |
| **Prevalent diabetes[5]** | 151 (14) | 481 (19) | 382 (33) | 206 (13) | 479 (17) | 354 (30) |
| **DHD-CVD index score[6]** | 89.2 ± 15.1 | 90.1 ± 14.7 | 87.6 ± 14.5 | N/A | N/A | N/A |
| **Self-rated health** | | | | | | |
| Low or moderate | 259 (24) | 559 (28) | 357 (30) | N/A | N/A | N/A |

(*Continued*)

**Table 1.** (Continued)

| | Cohort study | | | | | |
|---|---|---|---|---|---|---|
| | Alpha Omega Cohort | | | UCC-SMART cohort | | |
| | (*n* = 4,837) | | | (*n* = 5,533) | | |
| Good | 828 (76) | 2,016 (72) | 818 (70) | N/A | N/A | N/A |

[1] Values are means ± SD for normally distributed variables, medians [IQRs] for skewed variables or *n* (%) for categorical or discrete variables

[2] In AOC, defined as 1) no activity, 2) light activity ($\leq$ 3 METs), 3) intermediate activity (moderate or vigorous activity; > 3 METs on >0 to <5 days per week), 4) high activity (moderate or vigorous activity; > 3 METs on >5 days per week), in UCC-SMART defined as quartiles of total physical activity from all activities in MET hours / week

[3] In AOC, defined based on calculated ethanol intake from an FFQ as: abstainers (0 g/d), light (>0–10 g/d in males and > 0–5 g/d in females), moderate (>10–30 g/d in males and >5–15 g/d in females) and heavy (>30 g/d in males and > 15 g/d in females, in UCC-SMART defined based on standard drinks (10 g ethanol) per week according to a general questionnaire as: abstainers (0 drinks), light (>0–10 drinks), moderate (11–20 drinks) and heavy (>20 drinks).

[4] In AOC, measured in a non-fasting state, measured in fasting state in UCC-SMART, measured in a fasting state.

[5] In AOC, defined as a self-reported physician diagnosis, use of antidiabetic medication or elevated plasma glucose. In SMART, defined as in AOC, except for self-reported physician diagnosis

[6] DHD-CVD-index score, Dutch Healthy Diet Cardiovascular Disease index.

Table 2 presents pooled HRs across categories of BMI and WC in relation to all-cause mortality and CVD mortality in the total pooled population, in males and females. HRs for all-cause mortality were 1.31 (95% CI: 1.11, 1.41) for obese (BMI $\geq$30 kg/m$^2$) males and 1.10 (95% CI: 0.92, 1.30) for obese females compared to a BMI 25–30 kg/m$^2$. For those with a BMI < 25 kg/m$^2$, the HR was 1.08 (95% CI: 0.97, 1.20) in males and 1.21 (95% CI: 0.81, 1.81) in females for all-cause mortality. For WC, HRs for all-cause mortality were 1.18 (95% CI:1.06, 1.30) in males and 1.31 (95% CI: 1.05, 1.64) in females for the highest vs. middle category of WC. The lowest WC category (< 94 cm for males, < 80 cm for females) was not associated with an increased all-cause mortality risk. Results for BMI and WC with CVD mortality were mostly in line with results for all-cause mortality, especially in males. In females, results for CVD mortality were attenuated.

The associations of BMI and WC with mortality risk remained roughly similar after adjustment for potential intermediate factors including prevalent diabetes, systolic blood pressure, LDL-cholesterol and hs-CRP (model 3). Associations of BMI and WC with all-cause and CVD mortality in the AOC and UCC-SMART separately can be found in S2 and S3 Tables.

## Subgroup analyses

Fig 3 depicts associations of BMI with all-cause mortality in the total pooled population in subgroups. We found consistent U-shaped associations between BMI and all-cause mortality across subgroups of educational level, prevalent diabetes, diet quality and self-rated health. A higher risk of all-cause mortality was consistently observed across subgroups of patients with a BMI $\geq$30 kg/m$^2$ compared to those with BMI 25–30 kg/m$^2$, especially in patients with a low physical activity level and high educational level. A higher mortality risk was less pronounced for obese patients with low or moderate self-rated health, in whom the highest mortality risk was observed for a BMI <25 kg/m$^2$. For never smokers and patients with high physical activity, a BMI <25 kg/m$^2$ was not associated with a higher mortality risk. A higher WC was consistently associated with a higher all-cause mortality risk compared to the middle category of WC (S3 Fig) especially in those with diabetes and high physical activity. A low WC was not significantly associated with mortality except for those with diabetes and low or moderate self-rated

## All−cause mortality

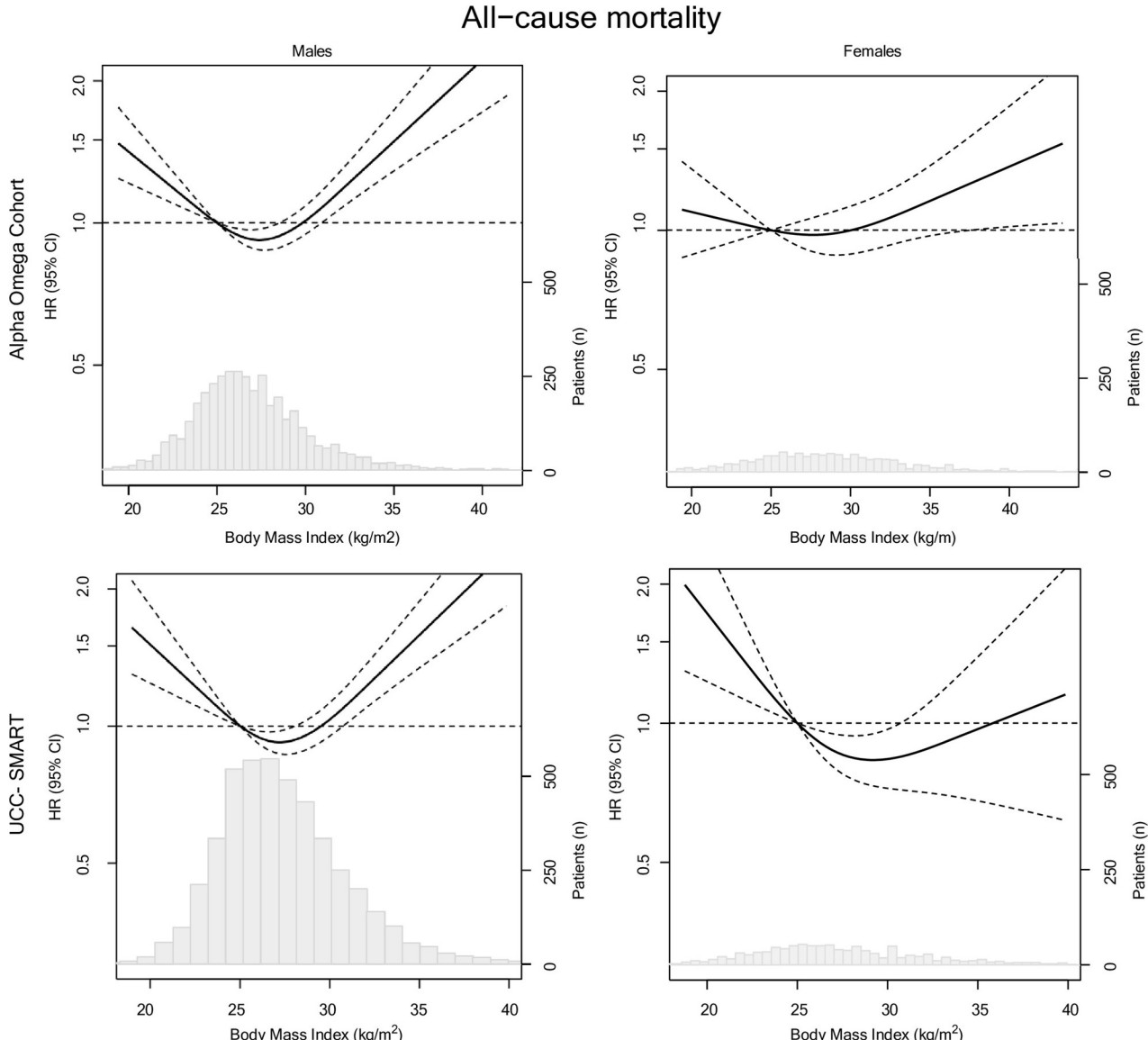

**Fig 1. Associations for BMI in relation to all-cause mortality in 4,837 CAD patients from AOC and 5,533 from the UCC-SMART stratified for sex.** Lines are restricted cubic splines, showing 3 knots at the 10[th], 50[th] and 90[th] percentiles. The y-axis shows the predicted HRs for mortality for any value of BMI, compared to the reference, set at 25 kg/m[2]. Results are presented for model 2.

health showing a trend towards an increased mortality risk with low WC in these groups. Results for subgroup analyses further stratified for sex can be found in S4–S7 Figs.

### Sensitivity analyses

Results for BMI and WC with all-cause mortality and CVD mortality remained robust after excluding patients with cancer at baseline (only performed in AOC) and after excluding the first 2 and 5 years of follow-up (S4–S9 Tables). Excluding patients with underweight (<0.5% of the study population) did not essentially change the results (data not shown).

## All−cause mortality

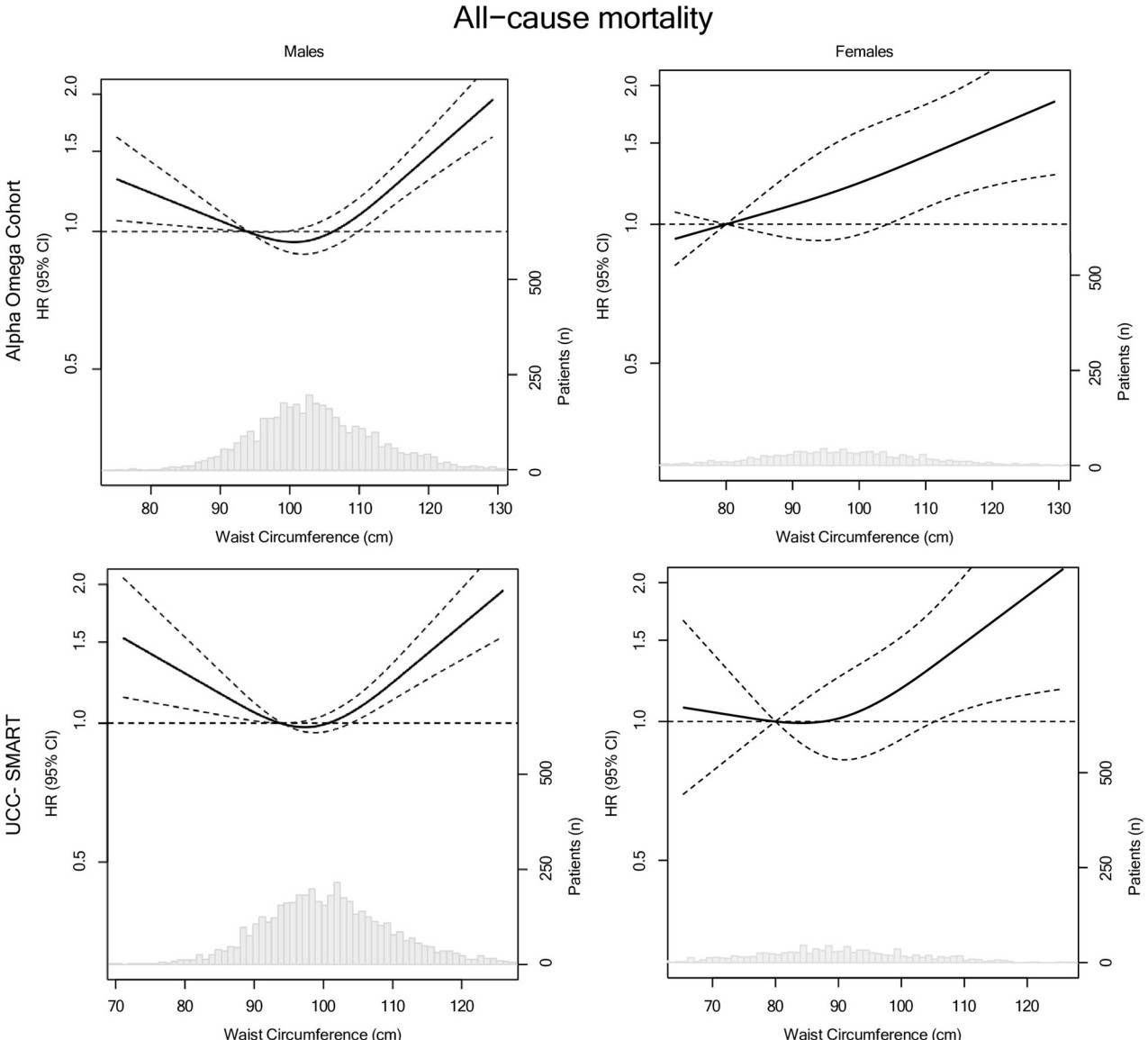

**Fig 2. Associations for WC in relation to all-cause mortality in 4,837 CAD patients from AOC and 5,533 from the UCC-SMART stratified for sex.** Lines are restricted cubic splines, showing 3 knots at the 10th, 50th and 90th percentiles. The y-axis shows the predicted HRs for mortality for any value of WC, compared to the reference, set at 94 cm for males and 80 cm for females. Results are presented for model 2.

## Discussion

In this pooled analysis of two cohorts of CAD patients, U-shaped relationships were found for BMI and long-term mortality risk, with a $\sim$10% lower mortality risk for a BMI of $\sim$27 kg/m$^2$ compared to those with a BMI of 25 kg/m$^2$. Results for WC followed a similar patten, showing the lowest mortality risk in those with an intermediate WC. U-shaped associations were observed across most subgroups of lifestyle and health determinants.

We observed non-linear associations between BMI and mortality with the lowest risk of all-cause mortality for a BMI of $\sim$27 kg/m$^2$ for both males and females. These non-linear findings were in line with a meta-analysis of 89 studies in 1.3 million CAD patients which also showed a lower risk of long-term mortality in patients that were overweight (RR: 0.78, 95% CI: 0.74,

**Table 2. Pooled hazard ratios (95%CI) for BMI and WC in relation to all-cause mortality and CVD mortality in 10,370 CAD patients from AOC and the UCC-SMART.**

| | | Body Mass Index | | | Waist Circumference | |
|---|---|---|---|---|---|---|
| | | Categories of BMI | | | Categories of WC | |
| | 1 \| BMI < 25 | 2 \| BMI ≥ 25–30 | 3 \| BMI ≥ 30 | 1 \| Males: WC < 94; Females: WC < 80 | 2 \| Males: WC ≥ 94–102; Females: WC ≥ 80–88 | 3 \| Males: WC ≥ 102; Females: WC ≥ 88 |
| **Total population** | | | | | | |
| n | 2,664 | 5,346 | 2,360 | 2,246 | 3,002 | 5,122 |
| Median | 23.7 | 27.2 | 32.2 | Males: 90 | Males: 98 | Males: 108 |
| | | | | Females: 75 | Females: 85 | Females: 98 |
| Person-years | 26,475 | 55,814 | 23,803 | 22,661 | 31,564 | 51,858 |
| **All-cause mortality** | | | | | | |
| Events | 919 | 1,791 | 843 | 600 | 930 | 2,024 |
| Crude model | 1.15 (1.06, 1.24)[1] | 1 | 1.10 (1.02, 1.19) | 1.00 (0.90, 1.11) | 1 | 1.27 (1.15, 1.40) |
| Model 1[2] | 1.10 (1.02, 1.19) | 1 | 1.26 (1.16, 1.37) | 1.04 (0.94, 1.15) | 1 | 1.29 (1.14, 1.45) |
| Model 2[3] | 1.10 (1.01, 1.19) | 1 | 1.23 (1.13, 1.33) | 1.04 (0.94, 1.16) | 1 | 1.19 (1.10, 1.29) |
| Model 3[4] | 1.12 (1.03, 1.21) | 1 | 1.17 (1.08, 1.27) | 1.06 (0.96, 1.18) | 1 | 1.16 (1.07, 1.26) |
| **CVD mortality** | | | | | | |
| Events | 412 | 808 | 308 | 282 | 405 | 933 |
| Crude model | 1.13 (1.00, 1.27) | 1 | 1.15 (1.03, 1.30) | 1.07 (0.83, 1.37) | 1 | 1.36 (1.20, 1.52) |
| Model 1 | 1.07 (0.95, 1.21) | 1 | 1.33 (1.18, 1.50) | 1.11 (0.91, 1.34) | 1 | 1.36 (1.21, 1.53) |
| Model 2 | 1.08 (0.96, 1.22) | 1 | 1.27 (1.13, 1.44) | 1.11 (0.96, 1.30) | 1 | 1.26 (1.12, 1.42) |
| Model 3 | 1.09 (0.98, 1.21) | 1 | 1.23 (1.09, 1.39) | 1.15 (0.99, 1.34) | 1 | 1.23 (1.09, 1.38) |
| **Males** | | | | | | |
| n | 2,083 | 4,484 | 1,703 | 1,963 | 2,640 | 3,649 |
| Median | 23.5 | 27.2 | 32.5 | 90 | 98 | 108 |
| Person-years | 2,0886 | 46,795 | 17,071 | 20,058 | 27,811 | 36,852 |
| **All-cause mortality** | | | | | | |
| Events | 716 | 1,488 | 591 | 541 | 837 | 1,417 |
| Crude model | 1.14 (1.03, 1.24) | 1 | 1.09 (0.99, 1.20) | 1.00 (0.89, 1.12) | 1 | 1.20 (0.94, 1.53) |
| Model 1 | 1.04 (0.90, 1.21) | 1 | 1.33 (1.21, 1.47) | 1.01 (0.91, 1.13) | 1 | 1.26 (1.11, 1.44) |
| Model 2 | 1.08 (0.97, 1.20) | 1 | 1.31 (1.11, 1.54) | 1.03 (0.93, 1.15) | 1 | 1.18 (1.06, 1.30) |
| Model 3 | 1.09 (0.99, 1.20) | 1 | 1.22 (1.11, 1.35) | 1.06 (0.94, 1.18) | 1 | 1.16 (1.03, 1.31) |
| **CVD mortality** | | | | | | |
| Events | 332 | 653 | 284 | 258 | 369 | 642 |
| Crude model | 1.17 (0.92, 1.50) | 1 | 1.20 (1.04, 1.38) | 1.06 (0.82, 1.36) | 1 | 1.29 (1.13, 1.47) |

*(Continued)*

**Table 2.** (Continued)

| | | Body Mass Index | | | Waist Circumference | |
|---|---|---|---|---|---|---|
| | | Categories of BMI | | | Categories of WC | |
| Model 1 | 1.07 (0.78, 1.48) | 1 | 1.46 (1.27, 1.69) | 1.07 (0.89, 1.30) | 1 | 1.29 (1.14, 1.47) |
| Model 2 | 1.10 (0.82, 1.46) | 1 | 1.39 (1.21, 1.60) | 1.10 (0.94, 1.30) | 1 | 1.22 (1.07, 1.39) |
| Model 3 | 1.12 (0.88, 1.45) | 1 | 1.35 (1.17, 1.55) | 1.14 (0.87, 1.33) | 1 | 1.20 (1.05, 1.37) |
| **Females** | | | | | | |
| n | 581 | 862 | 657 | 265 | 362 | 1473 |
| Median | 23.1 | 27.3 | 32.6 | 75 | 85 | 98 |
| Person-years | 5,588 | 9,018 | 6,732 | 2,570 | 3,762 | 15,005 |
| **All-cause mortality** | | | | | | |
| Events | 203 | 303 | 252 | 58 | 93 | 607 |
| Crude model | 1.22 (0.85, 1.77) | 1 | 1.08 (0.91, 1.28) | 1.05 (0.75, 1.46) | 1 | 1.46 (1.17, 1.83) |
| Model 1 | 1.28 (0.89, 1.85) | 1 | 1.11 (0.94, 1.31) | 1.25 (0.90, 1.73) | 1 | 1.45 (1.16, 1.82) |
| Model 2 | 1.21 (0.81, 1.81) | 1 | 1.10 (0.92, 1.30) | 1.10 (0.74, 1.64) | 1 | 1.31 (1.05, 1.64) |
| Model 3 | 1.26 (0.82, 1.95) | 1 | 1.04 (0.87, 1.25) | 1.18 (0.85, 1.67) | 1 | 1.26 (1.00, 1.58) |
| **CVD mortality** | | | | | | |
| Events | 80 | 155 | 116 | 88 | 36 | 291 |
| Crude model | 0.97 (0.46, 2.02) | 1 | 0.96 (0.76, 1.23) | 1.13 (0.67, 1.89) | 1 | 1.79 (1.08, 2.97) |
| Model 1 | 1.01 (0.47, 2.14) | 1 | 1.00 (0.78, 1.27) | 1.35 (0.80, 2.27) | 1 | 1.83 (1.29, 2.60) |
| Model 2 | 0.98 (0.44, 2.18) | 1 | 0.97 (0.75, 1.23) | 1.30 (0.77, 2.19) | 1 | 1.63 (1.14, 2.32) |
| Model 3 | 1.06 (0.46, 2.42) | 1 | 0.90 (0.70, 1.16) | 1.41 (0.83, 2.41) | 1 | 1.49 (1.04, 2.13) |

[1] Pooled hazard ratio (95% confidence interval) obtained from Cox proportional hazards models (all such values), using the middle category as the reference, and random effects meta-analysis

[2] Adjusted for age and sex, not adjusted for sex in sex-stratified results

[3] Adjusted as model 1, plus for smoking status, physical activity, educational level and alcohol intake, this model was used as the main model

[4] Adjusted as model 2, plus for diabetes, systolic blood pressure, LDL-cholesterol and hs-CRP.

0.82) compared to those with lower or higher BMIs [3, 29]. Current guidelines for the general population recommend a BMI between 18 and 25 kg/m$^2$ for optimal health, but the optimal BMI appears to be shifted to the right in CAD patients. Because of the observational nature of our study, we could not study underlying mechanisms to explain this paradox. However, physiologically, this discrepancy with better survival in CAD patients could be explained by the protective effects of metabolic reserves providing more resilience and protection in times of illness or stress [30, 31], especially during later stages of life as our population represents [32]. Methodologically, this discrepancy could possibly be explained by collider stratification bias [33], reverse causation [34] or confounding by smoking [35], as discussed below. Our

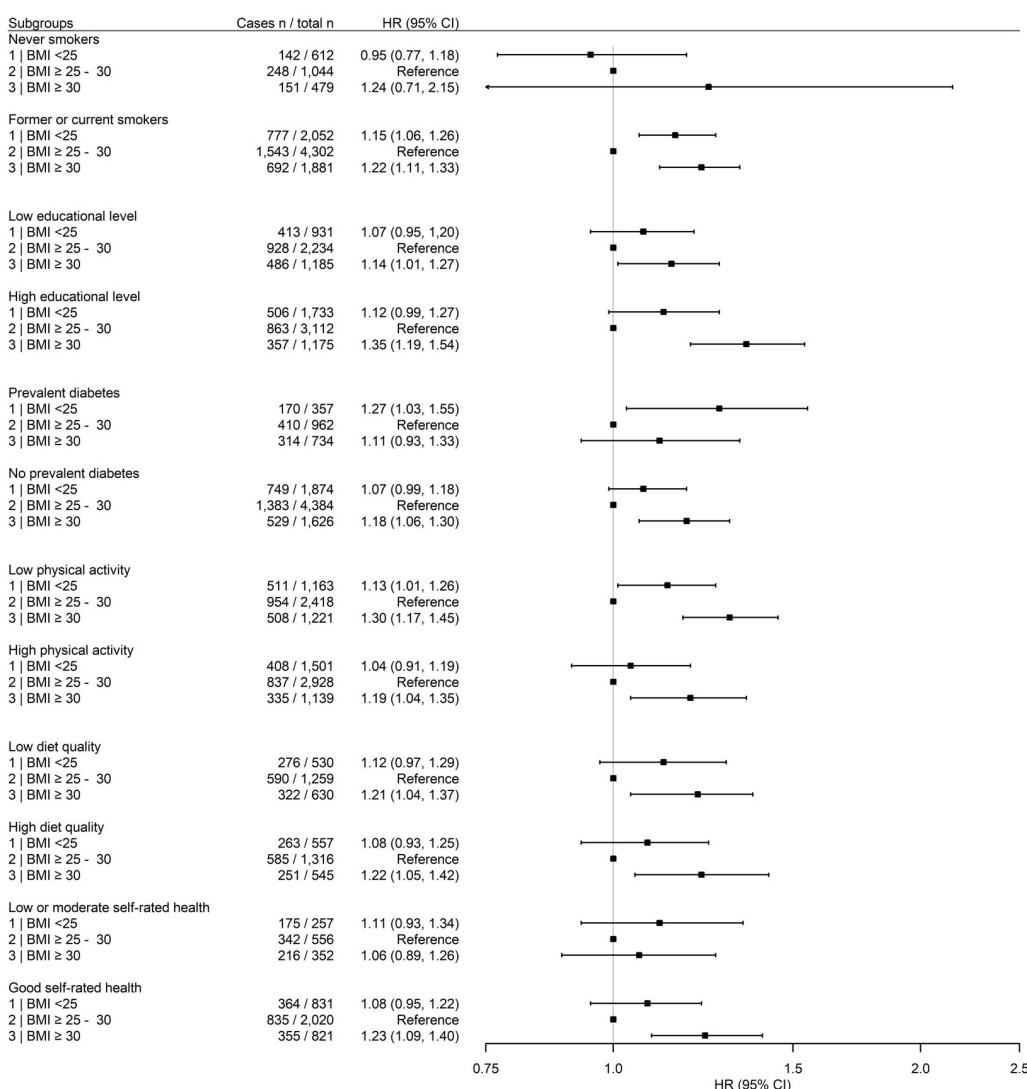

**Fig 3. Hazard ratios for BMI in relation to all-cause mortality in 10,370 CAD patients from AOC and the UCC-SMART, stratified for smoking status, educational level, diabetes, physical activity, diet quality and self-rated health.** Associations were adjusted according to variables in model 2 unless for variable stratified for. Results for diet quality and self-rated health only available from AOC.

results for patients with a BMI > 30 kg/m$^2$ were in accordance with the meta-analysis by Wang et al. [3] showing a higher risk of mortality, especially in those with a BMI > 35 kg/m$^2$ compared to patients with normal weight (BMI between 18.5–25.0 kg/m$^2$).

Non-linear associations were also found for WC with mortality risk in our pooled analysis of CAD patients, especially in males. Results for women were inconsistent across the separate cohorts but a high WC was consistently associated with an increased risk of mortality. The non-linear nature of the association between WC and mortality was also observed in meta-analyses including prospective studies in the general population [36, 37]. A Mendelian Randomization (MR) study showed an association between a higher genetically predicted WC and higher risk of CAD [38]. This MR study also showed higher risks for the cardiometabolic risk factors hypertension, diabetes and high cholesterol with increasing WC, which is specifically a marker of central obesity and is closely correlated with excessive visceral fat [39, 40]. Previous

intervention and observational studies also showed that excessive visceral fat was associated with pro-atherogenic properties such as inflammation, insulin resistance, higher LDL-cholesterol and other cardiometabolic risk factors such as elevated systolic blood pressure [39]. When our models were additionally adjusted for these factors (model 3), the associations for a high WC with mortality were attenuated but remained significant, indicating that the association can be partially explained by these variables, but not entirely.

We studied BMI and WC with mortality risk in subgroups of CAD patients that differed lifestyle factors and health status. In most strata, U-shaped associations were found, although some variation was observed. The HR for obesity, compared to BMI 25–30 kg/m$^2$, was 1.30 (95% CI: 1.17, 1.45) in patients with low physical activity and 1.19 (95% CI: 1.04, 1.35) in patients with high physical activity. Additionally, in patients with high physical activity, a BMI <25 kg/m$^2$ was no longer associated with a higher mortality risk compared to those with BMI 25–30 kg/m$^2$. Previous studies show that higher levels of physical activity and cardiorespiratory fitness can counterbalance the adverse effects of excess adiposity and accompanied disruptive metabolic functioning, which is substantiated to some extent by our results [41–43]. Irrespective of diet quality, a BMI between 25 and 30 kg/m$^2$ and an intermediate WC were associated with the lowest risk of mortality in our CAD patient population. Studies on healthy eating and diet quality show that the detrimental effects of obesity can be partly attenuated by a habitual high quality diet by improving the underlying cardiometabolic risk profile [44, 45]. In our study, obesity was more strongly related to all-cause mortality risk in patients with a high compared to low educational level (HR of 1.35 and 1.14, respectively, compared to BMI 25–30 kg/m$^2$), especially in male patients (HR of 1.46). A previous study in >85,000 US participants also showed a trend towards higher mortality risk in obese patients with a higher educational level compared to those with a lower educational level [46]. Educational level was in the current study used as a marker of SEP. SEP encompasses several factors including (health) inequalities, access to resources and issues related to privilege power and control [47] and a higher SEP was shown to be associated with lower obesity prevalence [48]. Unfortunately, we did not have information on all factors related to SEP; therefore, we used educational level instead. In conclusion, although, results might vary across subgroups, a BMI >30 kg/m$^2$ was consistently associated with a higher mortality risk in our CAD patient populations.

A higher mortality risk in patients with a BMI < 25 kg/m$^2$ and a low WC compared to intermediate BMI and WC values were observed in our study. The meta-analysis by Wang et al. [3] also showed higher mortality risks in underweight patients compared to those with normal weight. Especially for those with a low BMI, sources of bias should be noted in these type of analyses. Firstly, smoking is a well-known confounder in the association between BMI and mortality because it is associated with both a low body weight and an increased risk of mortality [49]. In our subgroup analysis in patients that never smoked, a BMI < 25 kg/m$^2$ compared to a BMI between 25 and 30 kg/m$^2$ was no longer associated with a higher risk of mortality, indicating that smoking modifies this association especially in those with a low BMI. Secondly, reverse causation bias leads to biased risk estimates due to pre-existing disease which could occur in our population of CAD patients. Underlying disease and possible worsening of the condition could lower BMI and increase the risk of mortality. Excluding the first two and five years of follow-up showed no indication that estimates were biased due to underlying disease. No modification of associations was also observed in subgroup analyses after exclusion of patients with diabetes and those with poor self-rated health and in a sensitivity analysis excluding patients with cancer at baseline ($n$ = 534). However, these sources of bias should still be considered and outlined when investigating associations between body anthropometrics and mortality in CAD patients.

Limitations of our study include the possible uncertain interpretation of the subgroup results due to smaller sample sizes, especially in females. Subgroup results were not adjusted for multiple testing thus these results should be interpreted with caution. They form the basis for further study rather than being statistical proof for group differences. With BMI as measure of adiposity, we were unable to discriminate between adipose tissue and lean body mass, and also age-related declines in muscle mass and wasting could lead to misinterpretation of BMI as measure of body adiposity. WC, on the contrary, better reflects central adiposity but results were comparable for both outcomes. While efforts were made to control for potential confounding variables, residual confounding may still be present, including confounding by diet quality and self-rated health that were only measured in the AOC. Further, certain variables, such as self-rated health and SEP were not fully captured due to limitations of the collected data. Findings of our study cannot be generalized to healthy populations and CAD patients from non-Caucasian origin, women were also underrepresented in our study.

Strengths of our study include the standardized BMI and WC measurements that were performed by research nurses as opposed to self-reported measurements that are often subject to bias [50]. In both cohorts we had extensive data on potential effect modifiers and confounders. Habitual diet in AOC was extensively assessed using a 203-item validated food frequency questionnaire. Standardized analyses were performed in two large cohorts of CAD patients from the Netherlands with long follow-up duration after data harmonization. RCS analysis was used to investigate the hypothesized non-linear associations for BMI and WC with mortality.

In summary, we found U-shaped associations for BMI and WC with long-term CVD and all-cause mortality risk in CAD patients. Patients with obesity and a large WC were at increased mortality risk, as were underweight patients, while mildly overweight patients (BMI $\sim$ 27 kg/m$^2$) had the lowest mortality risk. These U-shaped associations persisted across diverse subgroups of CAD patients characterized by varying lifestyle factors and health status. Our findings may have implications for risk stratification and effective risk communication to CAD patients.

## Supporting information

**S1 Table. Baseline characteristics of 10,370 patients in the Alpha Omega Cohort & UCC-S-MART, stratified for WC.**
(DOCX)

**S2 Table. Hazard ratios for BMI in relation to all-cause mortality and CVD mortality in 10,370 CAD patients from AOC and the UCC-SMART.**
(DOCX)

**S3 Table. Hazard ratios for WC in relation to all-cause mortality and CVD mortality in 10,370 CAD patients from AOC and the UCC-SMART.**
(DOCX)

**S4 Table. Hazard ratios for BMI in relation to all-cause mortality and CVD mortality in 9,620 CAD patients from AOC and the UCC-SMART excluding the first two years of follow-up.**
(DOCX)

**S5 Table. Hazard ratios for BMI in relation to CVD and all-cause mortality in 8,337 CAD patients from AOC and the UCC-SMART excluding the first five years of follow-up.**
(DOCX)

**S6 Table. Hazard ratios for WC in relation to all-cause mortality and CVD mortality in 9,620 CAD patients from AOC and the UCC-SMART excluding the first two years of follow-up.**
(DOCX)

**S7 Table. Hazard ratios for WC in relation to all-cause mortality and CVD mortality in 8,337 CAD patients from AOC and the UCC-SMART excluding the first five years of follow-up.**
(DOCX)

**S8 Table. Hazard ratios for BMI in relation to all-cause mortality and CVD mortality in 4,837 CAD patients from AOC excluding patients with cancer.**
(DOCX)

**S9 Table. Hazard ratios for WC in relation to all-cause mortality and CVD mortality in 4,837 CAD patients from AOC excluding patients with cancer.**
(DOCX)

**S1 Fig. Associations for BMI in relation to CVD mortality in 4,837 CAD patients from AOC and 5,533 from the UCC-SMART stratified for gender.** Lines are restricted cubic splines, showing 3 knots at the 10th, 50th and 90th percentiles. The y-axis shows the predicted HRs for mortality for any value of BMI, compared to the reference, set at 25 kg/m$^2$. Results are presented for model 2.
(TIF)

**S2 Fig. Associations for WC in relation to CVD mortality in 4,837 CAD patients from AOC and 5,533 from the UCC-SMART stratified for gender.** Lines are restricted cubic splines, showing 3 knots at the 10th, 50th and 90th percentiles. The y-axis shows the predicted HRs for mortality for any value of WC, compared to the reference, set at 94 cm for males and 80 cm for females. Results are presented for model 2.
(TIF)

**S3 Fig. Hazard ratios for WC in relation to all-cause mortality in 10,370 CAD patients from AOC and the UCC-SMART, stratified for smoking status, educational level, diabetes, physical activity, diet quality and self-rated health.** Associations were adjusted according to variables in model 2 unless for variable stratified for. Results for diet quality and self-rated health only available from AOC.
(TIF)

**S4 Fig. azard ratios for BMI in relation to all-cause mortality in 8,270 male CAD patients from AOC and the UCC-SMART, stratified for smoking status, educational level, diabetes and physical activity.** Associations were adjusted according to variables in model 2 unless for variable stratified for. Results for diet quality and self-rated health only available from AOC.
(TIF)

**S5 Fig. Hazard ratios for BMI in relation to all-cause mortality in 2,100 female CAD patients from AOC and the UCC-SMART, stratified for smoking status, educational level, diabetes and physical activity.** Associations were adjusted according to variables in model 2 unless for variable stratified for. Results for diet quality and self-rated health only available from AOC.
(TIF)

**S6 Fig. Hazard ratios for WC in relation to all-cause mortality in 8,270 male CAD patients from AOC and the UCC-SMART, stratified for smoking status, educational level, diabetes and physical activity.** Associations were adjusted according to variables in model 2 unless for variable stratified for. Results for diet quality and self-rated health only available from AOC. (TIF)

**S7 Fig. Hazard ratios for WC in relation to all-cause mortality in 2,100 female CAD patients from AOC and the UCC-SMART, stratified for smoking status, educational level, diabetes and physical activity.** Associations were adjusted according to variables in model 2 unless for variable stratified for. Results for diet quality and self-rated health only available from AOC. (TIF)

## Acknowledgments

We gratefully acknowledge the contribution of members of the Alpha Omega Group: I. van Damme, J.M. Geleijnse (principle investigator), L. Heerkens, N. Khandpur, M.G. Jacobo Cejudo, K. Pertiwi, A.C. van Westing, Division of Human Nutrition and Health, Wageningen University & Research, Wageningen, The Netherlands and members of the Utrecht Cardiovascular Cohort-Second Manifestations of ARTerial disease Study group (UCC-SMART study group): M.J. Cramer, H.M. Nathoe and M.G. van de Meer (co-PI), Department of Cardiology; G.J. de Borst and M. Teraa (co-PI), Department of Vascular Surgery; M.L. Bots and M. van Smeden, Julius Center for Health Sciences and Primary Care; M.H. Emmelot-Vonk, Department of Geriatrics, P.A. de Jong, Department of Radiology; A.T. Lely, Department of Gynaecology and Obstetrics; N.P. van der Kaaij, Department of Cardiothoracic Surgery; L.J. Kappelle and Y.M. Ruigrok, Department of Neurology; M.C. Verhaar, Department of Nephrology & Hypertension; J.A.N. Dorresteijn (co-PI), F.L.J. Visseren (PI), Department of Vascular Medicine, UMC Utrecht.

## Author Contributions

**Conceptualization:** Esther Cruijsen, Renate M. Winkels, Johanna M. Geleijnse.

**Formal analysis:** Esther Cruijsen.

**Funding acquisition:** Frank L. J. Visseren, Johanna M. Geleijnse.

**Investigation:** Esther Cruijsen.

**Methodology:** Esther Cruijsen, Nadia E. Bonekamp, Charlotte Koopal.

**Project administration:** Esther Cruijsen.

**Resources:** Frank L. J. Visseren, Johanna M. Geleijnse.

**Supervision:** Frank L. J. Visseren, Johanna M. Geleijnse.

**Visualization:** Esther Cruijsen.

**Writing – original draft:** Esther Cruijsen.

**Writing – review & editing:** Nadia E. Bonekamp, Charlotte Koopal, Renate M. Winkels, Frank L. J. Visseren, Johanna M. Geleijnse.

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
