## [Decision Letter · Decision Letter 0]

2 Jan 2024

PONE-D-23-38603Effect of lifestyle factors, education and indicators of health status on the association of body anthropometrics with long-term mortality in 10,370 patients with coronary artery diseasePLOS ONE

Dear Dr. Cruijsen,

Thank you for submitting your manuscript to PLOS ONE. After careful consideration, we feel that it has merit but does not fully meet PLOS ONE’s publication criteria as it currently stands. Therefore, we invite you to submit a revised version of the manuscript that addresses the points raised during the review process.

As you can see, the reviewers have requested substantial revisions to your manuscript.  We are certainly willing to reconsider a revised submission, but please know that this is not preliminary acceptance of your paper.  When returning your revised manuscript, please be sure to include a point-by-point summary of the suggestions of the reviewers that specifies how and where in the text you have addressed the suggestions.

We look forward to receiving your revised manuscript.

Kind regards,

Ricardo Ney Oliveira Cobucci, Ph.D

Academic Editor

PLOS ONE

Journal Requirements:

"This work was supported by a grant from the Regio Deal Foodvalley (162135). The Alpha Omega Trial (2002–2009), from which this cohort study emerged, was supported by Netherlands Heart Foundation grant 200T401; NIH, National Heart, Lung, and Blood Institute, and Office of Dietary Supplements grant R01HL076200."

Additional Editor Comments:

Dear authors,

. We are certainly willing to reconsider a revised submission, but please know that this is not preliminary acceptance of your paper. When returning your revised manuscript, please be sure to include a point-by-point summary of the suggestions of the reviewers that specifies how and where in the text you have addressed the suggestions.

Reviewers' comments:

Reviewer's Responses to Questions

**Comments to the Author**

1. Is the manuscript technically sound, and do the data support the conclusions?

Reviewer #1: Partly

Reviewer #2: Partly

2. Has the statistical analysis been performed appropriately and rigorously? 

Reviewer #1: Yes

Reviewer #2: No

3. Have the authors made all data underlying the findings in their manuscript fully available?

Reviewer #1: Yes

Reviewer #2: No

4. Is the manuscript presented in an intelligible fashion and written in standard English?

Reviewer #1: Yes

Reviewer #2: Yes

5. Review Comments to the Author

Reviewer #1: This manuscript of a prospective study is interesting because the authors proposed to evaluate the association of BMI and WC with mortality in CAD patients as well as their modifying effects. This well-written study also adds to the literature of mortality risk factors in CAD patients from two large cohorts. However, there are some important concerns to discuss, as follows:

Title:

The tittle should be replaced “body anthropometrics” with “body mass index and waist circumference” to be more specific and do not mislead readers.

Methods:

How many patients with BMI <18.5 kg/m2 (underweight) were included? Patients with underweight and normal weight (BMI 18.5 – 25) could have a different association with mortality and could affect the results.

To categorize levels of alcohol intake, why the amount of alcohol intake in each level differed between cohorts? Authors may covert amount of alcohol intake in drink to gram.

Please clarify why sex was used as covariate and stratification factor. Was there any interaction between BMI and sex, and WC and sex?

Results and Discussion:

Page 11, Line 4: Unit (years) should be added after the numbers.

For clarity, the percentage of CVD death and all-cause death should be presented. If the percentage is highly different, please discuss in this point.

Please carefully interpret the results form Figure 3. Some results cannot be addressed when the mortality risks were not statistically significant.

Table 1:

P-values among 3 BMI groups of each variable should be informed.

Please correct Age (y) to Age, year and serum blood lipids to serum lipids

Figure 1:

From RCS plots, female patients with BMI 25-30 kg/m2 seems to have lower mortality risk in UCC-SMART from plot whereas BMI >37 kg/m2 seems to have higher mortality risk. Please discuss in this point.

Figure 3: P for interaction between subgroups of each lifestyle factor should be presented to confirm the difference between these subgroups.

Please be consistent with the word "participants" or "patients".

Reviewer #2: The study "Effect of lifestyle factors, education and indicators of health status on the association of body anthropometrics with long-term mortality in 10,370 patients with coronary artery disease" by E. Cruijsen et al. is theoretically and clinically important. However, several conceptual and methodological aspects of the paper should be clarified and the title and conclusions amended.

Introduction

The authors summarize: "The extent to which lifestyle behaviors or comorbidities such as diabetes might explain or modify the J- or U-shaped relationships between BMI and WC and mortality risk in CHD patients is not yet clear. The nature of the relationship might also differ between male and female patients (13) (Mohammadi et al. 2013 review: In the univariate analysis, the risk was increased in the fifth quintile (hazard ratio 1.22, 95% confidence interval 1.07-1.39) compared to the first. In the multivariable-adjusted analysis, the risk was increased in the fourth and fifth quintiles (hazard ratio 1.21, confidence interval 1.03-1.43 and hazard ratio 1.25, confidence interval 1.04-1.50, respectively). Sex-specific analyses showed similar associations in men, while U-shaped associations were observed in women and the body mass index analyses). The authors should clarify: What are the research hypotheses in this paper and control for multiple testing:

H1: There are J- or U-shaped associations between BMI and WC and the risk of death in CAD patients.

H2.there are gender-specific differences: linear correlations in men and U-shaped correlations in women

H3.lifestyle factors (smoking, physical activity, healthy diet) influence this association

H4.Socioeconomic status (SES)/education influences this association

H5.health status indicators affect this association

Methods

- 4 837 CAD patients from the Alpha Omega Cohort (AOC) and 5 533 patients from the Utrecht Cardiovascular Cohort - Secondary Manifestations of ARTerial disease (UCC-SMART) were included in the study between 1996 and 2020. AOC was a randomized controlled trial of omega-3 fatty acids during the first three years of follow-up, while UCC-SMART focused on the treatment of cardiovascular disease. The patients were between 18 and 79 years old. Patients with a clinical diagnosis of CAD (defined as myocardial infarction, percutaneous coronary intervention, coronary bypass surgery and angina pectoris) were included. The study sample was therefore heterogeneous and was originally selected for other studies.

It is not clear whether a pre-planned secondary analysis was conducted.

-Dividing the sample according to clinical criteria into BMI <25; BMI 25-<30 and BMI ≥ 30 kg/m2 and comparing these three groups is, in my opinion, not statistically suitable to prove hypotheses 1-5 (see above). You need the entire sample for the calculation.

-What are the lifestyle factor variables you want to examine in this study? Smoking, physical activity, alcohol consumption, diet quality?

- Education as a proxy variable for SES should be explained.

- Measuring self-related health with a single item is very narrow and not suitable to prove H5. The DHD-CVD score consists of 16 items.

- Further psychological measures of depression etc. are missing, as is the use of psychotropic drugs/antidepressants

- Fig. 3-7 in the appendix provide no information for the hypotheses and can be omitted.

Discussion

- The good performance of overweight men and women (BMI 25-30) is astonishing; what is the reason for this?

- And what are the reasons for the poor outcome of CAD patients with normal weight (BMI <25- >18)? This is important for treatment guidelines for dietary habits in CAD patients, especially because a healthy diet had no effect on all-cause and CVD mortality.

- What is the reason for the U-shaped relationships between BMI and WC and mortality risk in CHD patients?

- What are the reasons for the sex difference?

- Why does all-cause mortality show this U-shaped relationship, but CVD mortality does not? One might expect the opposite.

- Diabetes, hypertension, LDL cholesterol, CRP, history of MI or CABG, and heart failure (NYHA, EF) are important physical modifiers of CVD mortality and all-cause mortality and should be discussed in more detail for the association between obesity/overweight/not overweight/not obese and mortality

- The limitations of the study results (see above) should be discussed in more detail.

Conclusion

The authors' statement: "Whether overweight is causally associated with a lower risk of premature mortality in CAD patients remains uncertain due to the limitations of this observational study. Obesity prevention ..... should be promoted in all CAD patients" cannot be accepted as overweight has the lowest mortality risk and should not be treated. Grade II and III obesity in CAD patients can be treated.

The title of this paper: "Influence of lifestyle factors, education and indicators of health status on the association of anthropometric body measures with long-term mortality in ...patients with coronary heart disease" is misleading because the calculation of the data did not produce these results.

-

6. PLOS authors have the option to publish the peer review history of their article (what does this mean?). If published, this will include your full peer review and any attached files.

Reviewer #1: No

Reviewer #2: No

---

## [Author Response · Author response to Decision Letter 0]

22 Feb 2024

See the .pdf file in the attached file for a full response to reviewers including changes in the manuscript. 

Comments from the editor

Author’s reply: We carefully checked the style requirements and changed the manuscript accordingly. 

2. Did you know that depositing data in a repository is associated with up to a 25% citation advantage. If you’ve not already done so, consider depositing your raw data in a repository to ensure your work is read, appreciated and cited by the largest possible audience. 

Author’s reply: Due to ethical reasons, the data cannot be made publicly available. However, we are open to collaboration and data sharing with other researchers. See the data availability statement for more information. 

"This work was supported by a grant from the Regio Deal Foodvalley (162135). The Alpha Omega Trial (2002–2009), from which this cohort study emerged, was supported by Netherlands Heart Foundation grant 200T401; NIH, National Heart, Lung, and Blood Institute, and Office of Dietary Supplements grant R01HL076200."

Please state what role the funders took in the study. If the funders had no role, please state: ""The funders had no role in study design, data collection and analysis, decision to publish, or preparation of the manuscript."" If this statement is not correct you must amend it as needed. Please include this amended Role of Funder statement in your cover letter; we will change the online submission form on your behalf.

Author’s reply: We included the role of the funders in the financial disclosure: “The funders had no role in the study design, data collection and analysis, decision to publish or preparation of the manuscript.” 

4. Please include captions for your Supporting Information files at the end of your manuscript, and update any in-text citations to match accordingly.

Author’s reply: Captions for the Supporting Information files were included at the end of the manuscript and citations are matched accordingly. 

Reviewer 1 

This manuscript of a prospective study is interesting because the authors proposed to evaluate the association of BMI and WC with mortality in CAD patients as well as their modifying effects. This well-written study also adds to the literature of mortality risk factors in CAD patients from two large cohorts. However, there are some important concerns to discuss, as follows.

Thank you for carefully reading our manuscript and for the valuable feedback.

Title:

1. The tittle should be replaced “body anthropometrics” with “body mass index and waist circumference” to be more specific and do not mislead readers.

Author’s reply: We agree that the title could be more specific. We also incorporated the comments of the other reviewer and changed the title into the following: 

Methods:

2. Methods: How many patients with BMI <18.5 kg/m2 (underweight) were included? Patients with underweight and normal weight (BMI 18.5 – 25) could have a different association with mortality and could affect the results.

Author’s reply: Of our total sample, <0.5% was underweight. Our continuous analysis indeed showed a higher risk of mortality in patients with a low BMI. In categorical analyses we combined underweight and normal weight patients in the same category. Leaving out the small number of underweight patients from our categorical analyses did not essentially change the results. We have added this information to the Results section.

3. To categorize levels of alcohol intake, why the amount of alcohol intake in each level differed between cohorts? Authors may covert amount of alcohol intake in drink to gram.

Author’s reply: Alcohol intake was measured differently in the two studies. A food frequency questionnaire (FFQ) with detailed information on alcohol consumption (types, amounts) was used in the Alpha Omega Cohort. Based on that information, sex-specific categories were defined that showed strong associations with mortality (Cruijsen et al. Am J Clin Nutr 2022). In UCC-SMART, a general questionnaire was used that assessed alcohol consumption in standard drinks per weeks which was subsequently categorized. Alcohol was not the primary exposure of interest for the present study, but a confounder, and we decided to maintain the cohort-specific categorization that showed the best fit with the data, consistent with previous publications of both studies. 

4. Please clarify why sex was used as covariate and stratification factor. Was there any interaction between BMI and sex, and WC and sex?

Author’s reply: Body fat mass and fat distribution differ between men and women, as does mortality risk. In our combined analyses, we therefore adjusted for sex as a confounder. In addition, we performed stratification to obtain more insight in the role of sex, as a potential effect modifier. When designing the current study, we set an a priori goal to derive separate risk estimates for men and women, as well as other subgroups, independent of whether differences in risk estimates would be identified. There is lack of female-specific data in cardiovascular medicine. Our focus on group-specific risk assessment is crucial for the practical application of our findings, such as risk prediction models and communication to patients, which is why we do not formally test for (statistical) interaction. Results from subgroup analyses should, however, be interpreted with caution. They form the basis for further study, rather than being statistical proof for group differences. This has now been emphasized in the Discussion section.

Results and Discussion:

5. Page 11, Line 4: Unit (years) should be added after the numbers.

Author’s reply: Done. 

6. For clarity, the percentage of CVD death and all-cause death should be presented. If the percentage is highly different, please discuss in this point.

Author’s reply: The percentage of deaths in the Alpha Omega Cohort (47%) is higher than in the UCC-SMART cohort (23%) but this was mainly due to the age of participants at study enrolment. Patients in the Alpha Omega Cohort were already 60-80 years and patients in UCC-SMART were 18-79 years. We added the percentages of total deaths to our Results section. 

7. Please carefully interpret the results from Figure 3. Some results cannot be addressed when the mortality risks were not statistically significant.

Author’s reply: We agree that these results should be interpreted with caution. We are not primarily interested in statistical significance, but rather in the magnitude of risk estimates and risk patterns across subgroups. We went through the Results section belonging to Figure 3 and adjusted the phrasing in several places, showing more uncertainty. 

Table 1:

8. P-values among 3 BMI groups of each variable should be informed.

Author’s reply: We adhered to STROBE principles (von Elm et al. PLOS Med 2007) and did not include p-values in the baseline table for the following reasons. 1) These p-values would only reflect univariate associations across one variable. However, most of the variables in the baseline table will be biased by other variables (e.g. sex and age) of the patients. 2) Statistically significant differences (two-sided P <0.05) in baseline characteristics are often used to select confounders. However, small differences in a confounder may still impact the associations and should also be adjusted for anyhow (Vandenbroucke et al. Ann Intern Med 2007). 

9. Please correct Age (y) to Age, year and serum blood lipids to serum lipids

Author’s reply: We agree, and this has been changed in Table 1. 

Figure 1:

10. From RCS plots, female patients with BMI 25-30 kg/m2 seems to have lower mortality risk in UCC-SMART from plot whereas BMI >37 kg/m2 seems to have higher mortality risk. Please discuss in this point.

Author’s reply: From the RCS plot, we can indeed observe that women from UCC-SMART with a BMI between 25-30 kg/m2 have the lowest mortality risk. However, caution is warranted in interpreting results for BMI >30 kg/m2 onwards, as the 95% confidence interval includes the value of 1.00. It appears that those with BMI >37 kg/m2 may have a higher risk, but there is a large uncertainty around that estimate and we are hesitant to draw conclusions. 

Figure 3:

11. P for interaction between subgroups of each lifestyle factor should be presented to confirm the difference between these subgroups.

Author’s reply: As mentioned in our response to comment 4, our objective in this study was to obtain separate risk estimates within subgroups in order to focus on the practical application of our results in risk prediction models and communication to patients. Therefore we do not formally test for (statistical) interactions. Furthermore, more insight in potential effect measure modification can be obtained when the magnitude of the associations is considered, rather than evaluating a p-values. 

12. Please be consistent with the word "participants" or "patients".

Author’s reply: We consistently changed all wording of “participants” into “patients”. 

Reviewer 2 

The study "Effect of lifestyle factors, education and indicators of health status on the association of body anthropometrics with long-term mortality in 10,370 patients with coronary artery disease" by E. Cruijsen et al. is theoretically and clinically important. However, several conceptual and methodological aspects of the paper should be clarified and the title and conclusions amended.

Thank you for carefully reading our manuscript and for the valuable feedback. 

Introduction

1. The authors summarize: "The extent to which lifestyle behaviors or comorbidities such as diabetes might explain or modify the J- or U-shaped relationships between BMI and WC and mortality risk in CHD patients is not yet clear. The nature of the relationship might also differ between male and female patients (13) (Mohammadi et al. 2013 review: In the univariate analysis, the risk was increased in the fifth quintile (hazard ratio 1.22, 95% confidence interval 1.07-1.39) compared to the first. In the multivariable-adjusted analysis, the risk was increased in the fourth and fifth quintiles (hazard ratio 1.21, confidence interval 1.03-1.43 and hazard ratio 1.25, confidence interval 1.04-1.50, respectively). Sex-specific analyses showed similar associations in men, while U-shaped associations were observed in women and the body mass index analyses). The authors should clarify: What are the research hypotheses in this paper and control for multiple testing:

H1: There are J- or U-shaped associations between BMI and WC and the risk of death in CAD patients.

H2.there are gender-specific differences: linear correlations in men and U-shaped correlations in women

H3.lifestyle factors (smoking, physical activity, healthy diet) influence this association

H4.Socioeconomic status (SES)/education influences this association

H5.health status indicators affect this association

Author’s reply: We agree with your line of reasoning and the hypotheses that were tested in this manuscript, in particular H1 and H2. However, our study of potential modifications by lifestyle and health determinants (H3, H4, H5) should be considered as exploratory. Insight in these modifications can be useful for risk stratification, risk prediction models and communication to patients. We did not define hypothesis for stratified analyses because we were only interested in the associations and risk patterns across subgroups. With respect to multiple testing, the results of our subgroup analyses form the basis for further study rather than being a statistical proof for group differences. We have rephrased several sentences in our Introduction and Discussion section.

Methods

2. 4 837 CAD patients from the Alpha Omega Cohort (AOC) and 5 533 patients from the Utrecht Cardiovascular Cohort - Secondary Manifestations of ARTerial disease (UCC-SMART) were included in the study between 1996 and 2020. AOC was a randomized controlled trial of omega-3 fatty acids during the first three years of follow-up, while UCC-SMART focused on the treatment of cardiovascular disease. The patients were between 18 and 79 years old. Patients with a clinical diagnosis of CAD (defined as myocardial infarction, percutaneous coronary intervention, coronary bypass surgery and angina pectoris) were included. The study sample was therefore heterogeneous and was originally selected for other studies.

It is not clear whether a pre-planned secondary analysis was conducted.

Author’s reply: All analyses were based on existing observational data. Regarding the Alpha Omega Cohort, patients had been randomized to low doses of omega-3 fatty acids during the first three years of follow-up. The intervention did not impact the risk of major cardiovascular events, and can be ignored as potential confounder (due to randomization). After the intervention, patients were followed for cause-specific mortality using national registers. We used a predefined analysis plan for this pooled cohort analysis, with the primary aim to study the association between BMI/WC and mortality and the potential modifying role of lifestyle and healthy determinants. Analyses like presented here are the main purpose of these observational cohort studies. 

3. Dividing the sample according to clinical criteria into BMI <25; BMI 25-<30 and BMI ≥ 30 kg/m2 and comparing these three groups is, in my opinion, not statistically suitable to prove hypotheses 1-5 (see above). You need the entire sample for the calculation.

Author’s reply: We agree that making use of the total study population for each analysis is preferred. For testing hypothesis 1 and 2 we used the entire sample in our restricted cubic splines analysis. Since this continuous analysis revealed non-linear associations, we cannot perform analysis in subgroups per increment in BMI or WC, assuming linearity of the association. To warrant comparability we decided to perform categorical analyses for the entire sample and across subgroups. 

4. What are the lifestyle factor variables you want to examine in this study? Smoking, physical activity, alcohol consumption, diet quality?

Author’s reply: Yes, indeed, these are the lifestyle variables that we included in our study. We focused on the upstream, modifiable determinants of BMI and waist circumference, which are relevant for prevention and personalized risk communication to patients. Other important lifestyle factors such as stress and sleep might also be relevant but both cohort studies lacked information on this. 

5. Education as a proxy variable for SES should be explained.

Author’s reply: We operationalized SES (also referred to SEP: socioeconomic position) as the highest level of education achieved. This was assessed through a questionnaire containing seven response categories, which were later aggregated into four categories for analysis (see Methods). We realize that the variable ‘education’ is only a proxy, since it does not cover all factors related to SEP, such as income and (former) occupation. Unfortunately, we did not have information on the latter in our cohort. 

We have added the following information to our Methods and Discussion. 

6. Measuring self-related health with a single item is very narrow and not suitable to prove H5. The DHD-CVD score consists of 16 items.

Author’s reply: Even though self-rated health was assessed with one question, it showed significant predictive power for preterm mortality in our cohort of CAD patients. We agree that this variable does not capture the entirety of a patient’s health. We have replaced “indicators of health status” with “lifestyle and health determinants”. Alongside self-rated health, this denominator includes the presence of diabetes and lifestyle factors including smoking, physical activity and healthy diet. 

We used the DHD-CVD index as measure of adherence to 16 dietary guidelines for cardiovascular patients, as formulated by the Health Council of the Netherlands (www.gr.nl/en). The index consists of 16 items which were summarized in a total score that was used for our analyses. 

7. F

---

## [Decision Letter · Decision Letter 1]

17 Mar 2024

PONE-D-23-38603R1Association of body mass index and waist circumference with long-term mortality risk in 10,370 coronary patients and potential modification by lifestyle and health determinantsPLOS ONE

Dear Dr. Cruijsen,

Thank you for submitting your manuscript to PLOS ONE. After careful consideration, we feel that it has merit but does not fully meet PLOS ONE’s publication criteria as it currently stands. Therefore, we invite you to submit a revised version of the manuscript that addresses the points raised during the review process.

There are also suggestions from reviewers that must be included in the manuscript, so that together with the reviewers we can reassess whether the quality will allow us to recommend publication.

We look forward to receiving your revised manuscript.

Kind regards,

Ricardo Ney Oliveira Cobucci, Ph.D

Academic Editor

PLOS ONE

Journal Requirements:

Reviewers' comments:

Reviewer's Responses to Questions

**Comments to the Author**

1. If the authors have adequately addressed your comments raised in a previous round of review and you feel that this manuscript is now acceptable for publication, you may indicate that here to bypass the “Comments to the Author” section, enter your conflict of interest statement in the “Confidential to Editor” section, and submit your "Accept" recommendation.

Reviewer #2: (No Response)

2. Is the manuscript technically sound, and do the data support the conclusions?

Reviewer #2: Partly

3. Has the statistical analysis been performed appropriately and rigorously? 

Reviewer #2: (No Response)

4. Have the authors made all data underlying the findings in their manuscript fully available?

Reviewer #2: Yes

5. Is the manuscript presented in an intelligible fashion and written in standard English?

Reviewer #2: Yes

6. Review Comments to the Author

Reviewer #2: The manuscript is much better now. I have only two points where I am not satisfied with the authors' response:

1. the authors have cited the work of Mohammadi et al (2013) who performed a univariate analysis and found that the all-cause mortality risk was increased in the fifth quintile compared to the first quintile. In the multivariable-adjusted analysis, the risk was increased in the fourth and fifth quintiles. Why did the authors not use this type of calculation in each substudy separately and in the combined study to avoid the somewhat artificial clinical BMI classification?

2 The authors stated: "We did not define hypotheses ... because we were only interested in the associations and risk patterns between the subgroups" and "our study on possible changes by lifestyle and health factors (H3, H4, H5) should be considered exploratory."

My response: if they don't have research hypotheses, they shouldn't publish their data, but they agreed to my suggestion related to two hypotheses that could be answered with their data:

"H1: There are J- or U-shaped associations between BMI and WC and risk of death in CAD patients.

H2: There are gender differences: linear correlations in men and U-shaped correlations in women"

If the authors agree, exploratory research with less statistical power could describe these associations:

1.Lifestyle factors (smoking, physical activity, healthy diet) influence this association

2.socioeconomic status (SES)/education influences this association

3.health status indicators influence this relationship

This structure of the research paper should be given at the end of the introduction.

Minor

In my opinion, omitting several tables in the appendix would have made the structure of the paper clearer, but if the authors want to keep them in the paper, they should do so.

7. PLOS authors have the option to publish the peer review history of their article (what does this mean?). If published, this will include your full peer review and any attached files.

Reviewer #2: **Yes: **Hans-Christian Deter

---

## [Author Response · Author response to Decision Letter 1]

10 Apr 2024

Response to the reviewers 

Author’s reply: We carefully reviewed our reference list and it is complete and correct. We checked for any possible retracted papers using Retraction Watch, confirming that none of our included references has been retracted. 

Reviewer 2

The manuscript is much better now. I have only two points where I am not satisfied with the authors' response.

Author’s reply: Thank you for reviewing our revised manuscript. We hope to have adequately addressed your raised comments now. 

1. The authors have cited the work of Mohammadi et al (2013) who performed a univariate analysis and found that the all-cause mortality risk was increased in the fifth quintile compared to the first quintile. In the multivariable-adjusted analysis, the risk was increased in the fourth and fifth quintiles. Why did the authors not use this type of calculation in each substudy separately and in the combined study to avoid the somewhat artificial clinical BMI classification? 

Author’s reply: The article of Mohammadi et al (2020) used quintiles to categorise waist circumference and they used three groups (normal weight, overweight, obesity) to categorise BMI. Besides our clinical BMI and WC categorisation in three groups, we used restricted cubic splines to model continuous associations between BMI and WC with mortality risk. This method uses all available datapoints for BMI or WC rather than a categorisation of the data. By using these two methods, we believe we have adequately captured the (non-linear) associations between BMI and WC with mortality . 

2 The authors stated: "We did not define hypotheses ... because we were only interested in the associations and risk patterns between the subgroups" and "our study on possible changes by lifestyle and health factors (H3, H4, H5) should be considered exploratory."

My response: if they don't have research hypotheses, they shouldn't publish their data, but they agreed to my suggestion related to two hypotheses that could be answered with their data:

"H1: There are J- or U-shaped associations between BMI and WC and risk of death in CAD patients.

H2: There are gender differences: linear correlations in men and U-shaped correlations in women"

If the authors agree, exploratory research with less statistical power could describe these associations:

1.Lifestyle factors (smoking, physical activity, healthy diet) influence this association

2.socioeconomic status (SES)/education influences this association

3.health status indicators influence this relationship

This structure of the research paper should be given at the end of the introduction.

Author’s reply: We agree that our analyses on lifestyle factors, educational level and health status indicators should be regarded as exploratory. We revised the sentence outlining the structure of the manuscript to emphasize that these analyses are indeed exploratory.

Based on comments from other reviewers, we redefined our exploratory factors as lifestyle and health determinants, and we delineate all determinants in our introduction. 

The new sentence in line 65 is as follows: Exploratory analyses were performed to obtain insight in potential effect modification of these associations by lifestyle and health determinants including smoking, physical activity, diet quality, educational level, diabetes and self-rated health.

3. In my opinion, omitting several tables in the appendix would have made the structure of the paper clearer, but if the authors want to keep them in the paper, they should do so.

Author’s reply: As mentioned in our previous response, we are indeed in favour of including the supplementary tables for risk stratification, communication and future meta-analyses.

---

## [Decision Letter · Decision Letter 2]

24 Apr 2024

Association of body mass index and waist circumference with long-term mortality risk in 10,370 coronary patients and potential modification by lifestyle and health determinants

PONE-D-23-38603R2

Dear Dr. Cruijsen,

We’re pleased to inform you that your manuscript has been judged scientifically suitable for publication and will be formally accepted for publication once it meets all outstanding technical requirements.

Kind regards,

Ricardo Ney Oliveira Cobucci, Ph.D

Academic Editor

PLOS ONE

Additional Editor Comments (optional):

Reviewers' comments:

Reviewer's Responses to Questions

**Comments to the Author**

1. If the authors have adequately addressed your comments raised in a previous round of review and you feel that this manuscript is now acceptable for publication, you may indicate that here to bypass the “Comments to the Author” section, enter your conflict of interest statement in the “Confidential to Editor” section, and submit your "Accept" recommendation.

Reviewer #2: All comments have been addressed

2. Is the manuscript technically sound, and do the data support the conclusions?

Reviewer #2: Yes

3. Has the statistical analysis been performed appropriately and rigorously? 

Reviewer #2: Yes

4. Have the authors made all data underlying the findings in their manuscript fully available?

Reviewer #2: Yes

5. Is the manuscript presented in an intelligible fashion and written in standard English?

Reviewer #2: Yes

6. Review Comments to the Author

Reviewer #2: (No Response)

7. PLOS authors have the option to publish the peer review history of their article (what does this mean?). If published, this will include your full peer review and any attached files.

Reviewer #2: No

---

## [Editor Report · Acceptance letter]

7 May 2024

PONE-D-23-38603R2 

PLOS ONE

Dear Dr. Cruijsen, 

I'm pleased to inform you that your manuscript has been deemed suitable for publication in PLOS ONE. Congratulations! Your manuscript is now being handed over to our production team.

Kind regards, 

on behalf of

PROFESSOR Ricardo Ney Oliveira Cobucci 

Academic Editor

PLOS ONE